# The Effects of Fluidized Bed Drying, Soaking, and Microwaving on the Phytic Acid Content, Protein Structure, and Digestibility of Dehulled Faba Beans

Shu Cheng [1,*], Daniel J. Skylas [2], Chris Whiteway [2], Valeria Messina [1] and Timothy A. G. Langrish [1,*]

[1] School of Chemical and Biomolecular Engineering, The University of Sydney, Camperdown, NSW 2006, Australia; valeria.messina@sydney.edu.au

[2] Australian Export Grains Innovation Centre, North Ryde, NSW 2113, Australia; daniel.skylas@aegic.org.au (D.J.S.); chris.whiteway@aegic.org.au (C.W.)

[*] Correspondence: sche2348@uni.sydney.edu.au (S.C.); timothy.langrish@sydney.edu.au (T.A.G.L.)

**Abstract:** Different pre-treatments of pulses affect the content of antinutritional factors and protein digestibility. This study addresses the challenge of removing phytic acid, which is one of the major anti-nutritional factors present in faba beans. From this study, fluidized bed drying at 120 °C and 140 °C removed 8–22% of the phytic acid present. Phytic acid is thermally stable, and drying did not lead to a large reduction in concentration. Greater drying temperatures and drying times had little effect on the removal of phytic acid. Soaking the dehulled faba beans in 0.1% citric acid for 12 h at 37 °C removed 51 ± 11% of the phytic acid. After soaking for 12 h, microwaving the faba beans for two minutes removed over 70% of the phytic acid, including soaking in water or soaking in 0.1% citric acid solution. The mechanism for phytic acid reduction after microwaving appears to be linked with changes in the cotyledon cellular structure of the faba bean, as demonstrated using scanning electron microscopy. The protein secondary structure in the faba bean was modified after microwaving. The in vitro protein digestibility of dehulled faba beans used in this study increased by 15.7% compared with the fresh faba beans of 75.5 ± 0.5%. The in vitro digestibility of dehulled faba beans increased to 88.3 ± 0.6% after two minutes of microwaving, so both dehulling and microwaving improved the digestibility of the faba bean proteins.

**Keywords:** pulse protein; phytic acid; dehulling; microwaving; fluidized bed drying

## 1. Introduction

Faba beans are an important crop that is grown and consumed around the world [1]. In Australia, the average annual faba bean production from 2016–2021 was 391,000 t [2]. Pulses, such as faba beans, are high in proteins and nutritional value, containing dietary fibre, vitamins, minerals, and phytochemicals [3]. However, the digestibility and bioavailability of nutrients are reduced by the presence of anti-nutritional components [4]. Pulses are consumed and cooked, whole or processed, as dehulled splits, flours, and ingredients, and this processing can reduce the anti-nutritional contents [4,5]. The objective of this study was to remove anti-nutritional factors, particularly phytic acid, from faba beans with different treatments. In faba beans, a-amylase inhibitor, haemagglutinin activity, and phytic acid are the three most abundant anti-nutritional factors. Phytic acid, also known as nositol hexaphosphate, serves as the main form of storage of phosphorus in the seed. Phytic acid creates complexes with minerals, especially essential minerals such as iron, zinc, magnesium, and calcium, exhibiting "chelation". This chelation limits the bioavailability of these minerals in the human diet. This situation can lead to significant mineral deficiencies in humans and animals [6]. Kumar et al. [7] compared the anti-nutritional factor contents in different types of pulses (green gram, chickpeas, pigeon pea, black gram, filed pea, faba beans, and cowpea). The content of phytic acid in faba beans is in the range of

6.4–21.7 mg/g. Compared with chickpeas and lentils, the content of phytic acid in faba beans is higher. This is also an important reason to study the removal of phytic acid from faba beans. This study not only aimed to examine the influence of microwaving on phytic acid levels in faba beans but also studied the changes in the protein structure of faba beans and the changes in the cell structure of faba beans during microwaving.

Fluidized bed drying is a common industry practice. Fluidized bed drying of solids has become increasingly important in food, catalyst, and pharmaceutical processing due to the high heat and mass transfer rates [8,9]. In a previous study [10], a drying schedule was applied to the fluidized bed drying of chickpeas, which not only minimized the denaturation of the secondary structure of the protein during the drying process but also greatly reduced the content of trypsin inhibitors in the chickpeas. From Pande and Mishra [11], fluidized bed drying the green gram seed at a low temperature (50–70 °C) has no significant effects on the phytic acid levels. Therefore, in this study, the fluidized bed drying process at different and higher temperatures was applied to reduce the content of phytic acid in the faba beans.

Thermal treatments, such as cooking, have previously been reported to deactivate heat-sensitive factors such as trypsin inhibitors and volatile compounds [12]. Thermal treatments have been reported to have no significant effects on the removal of phytic acid. Therefore, other potential treatments were investigated. Table 1 from Sarkhel et al. [13], summarizes the experimental results of different pre-treatments for different pulse-type removals of phytic acid. For lentils, soaking under acidic conditions also removed 37% of the phytic acid [13]. Soaking Indian tribal pulses under alkaline conditions removed 11% of the phytic acid [14], so this study also investigated the effects of different soaking conditions on the removal of phytic acid from faba beans.

Microwaving is another standard industrial practice. It is often used in timber drying to change product properties and in food drying to reduce anti-nutritional factors. In the review by Sarkhel et al. [13], Table 1 shows that microwave drying after soaking can remove around 50% of the phytic acid [15] in lentils. Suhag et al. [16] used microwave processing to reduce anti-nutritional factors, such as phytic acid, trypsin inhibitors, tannins, saponins, and oxalates in pulses. This approach also improves the safety and quality of edible grains. The advantage of microwave treatment technology lies in the rapid, safe, and green reduction of anti-nutritional factors. According to Rafiq et al. [17], the power and duration of microwave treatment had a significant effect on the inactivation of phytic acid. Most of this research was aimed at the reduction of phytic acid in chickpeas [18], soybeans [19], and some other grains [20,21]. Few studies have investigated the effects of the microwaving process on phytic acid in faba beans, in conjunction with drying.

As drivers for sustainability increase, the importance of plant proteins is growing for food security [22]. According to the study from Kumar et al. [7], phytic acid can also form complexes with proteins, thereby reducing protein solubility. Phytic acid reduces the activity of key digestive enzymes such as lipase, a-amylase, pepsin, trypsin, and chymotrypsin [7]. In this study, in vitro protein digestibility was investigated after different treatment conditions. There is a range of phytic acid concentrations in pulses, depending on the type of pulse and the processing involved [23]. Therefore, it is important to reduce the content of phytic acid during processing. Table 1 shows the different amounts of reduction for phytic acid under different treatment conditions in the literature.

**Table 1.** Phytic acid reduction in pulses under different treatment conditions.

| Pulse | Processing Conditions | Phytic Acid Reduction (%) | References |
|---|---|---|---|
| Faba beans | Soaking | 32.7% | [23] |
| Lentils | Soaking in acidic condition | 37% | [15] |
| Indian tribal pulse | Soaking in alkaline condition | 11% | [14] |
| Lentils | Microwave + soaking | 45–52% | [15] |
| Green gram seed | Fluidized bed drying (50–70 °C) | Not significant | [11] |

According to Alonso et al. [23], soaking can reduce phytic acid levels by $30 \pm 3\%$. However, Shi et al. [24] found that there was no detectable decrease in phytic acid levels after soaking any of the seeds. Therefore, soaking may not be the best method to remove phytic acid. It may be affected by environmental factors (such as the temperature and pH) and has large variability. For example, Alonso et al. [23], Vidal-Valverde et al. [25], and Siddhuraju et al. [14] studied the use of different soaking conditions to remove phytic acid, finding that soaking the lentils under acidic conditions removed 37% of the phytic acid. According to Vidal-Valverde et al. [25], Indian tribal pulses soaked under alkaline conditions removed 11% of the phytic acid. According to Sharif et al. [15], microwaving is a potential method to remove phytic acid, where soaking and microwaving the pulses together can remove 45–52% of the total amount of phytic acid from lentils.

Here, we discuss the fluidized bed drying of dehulled faba beans first, including the moisture content–time curves and the temperature–time curves for dehulled faba beans and the amount of phytic acid reduction during the drying processes at 120 °C and 140 °C. Second, phytic acid reduction during the soaking and microwaving process is discussed. Third, the secondary protein structure and the physical structure of dehulled faba beans at different treatments are described, using Fourier transform infrared spectroscopy and scanning electron microscopy, respectively. Finally, in vitro protein digestibility from different treatment processes was studied.

## 2. Materials and Methods

The dehulled faba bean (*Vicia faba*) seed material used in this study was provided by the Australia Export Grains Innovation Centre (AEGIC) in Sydney.

### 2.1. Fluidized Bed Drying

Faba beans were dried using a fluidized bed dryer at two different temperature settings (120 and 140 °C). Figure 1 shows a schematic diagram showing the arrangement of the fluidized bed equipment. Both fans 1 and 2 work in tandem to control the air flow rate through the fluidized bed. The flow velocity was measured to be about 8.5 m·s$^{-1}$ at the fluidized bed by using an orifice-plate flowmeter. The faba beans were fluidized and sub-samples collected at two-minute intervals for analysis of the moisture content (oven-drying method). After weighing, the faba beans were placed in an 80 °C drying oven for a period of 24 h.

$$M = \frac{m_0 - m_1}{m_0} \times 100\% \tag{1}$$

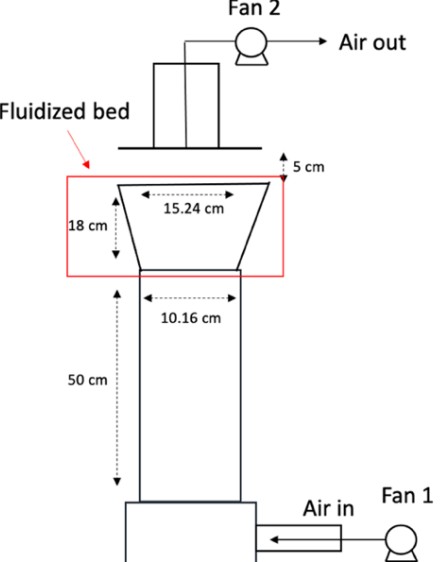

**Figure 1.** The fluidized bed used in this study.

Equation (1) was used to calculate the moisture content of faba beans. Here, $m_0$ is the mass of the faba beans before drying, and $m_1$ is the mass of faba beans after drying. The drying curve was obtained from the moisture contents measured as functions of time. Then, the drying curve was fitted by the equation '$X - C = A \exp(-B\,t)$', where $X$ is the fitted moisture content, $A$, $B$, and $C$ are fitted constants, and $t$ is the drying time.

Due to the rapid movements of the faba beans in the fluidized bed drying process, the actual faba bean temperatures are difficult to measure directly. To study the temperature of faba beans during the drying process, the following Equation (2) was used in this study, from Keey [26]:

$$f = \frac{T_G - T_s}{T_G - T_W} = \frac{-(dX/dt)}{-(dX/dt)_{max}} \tag{2}$$

where $T_G$ is the dry-bulb temperature, $T_W$ is the wet-bulb temperature, and $T_s$ is the temperature of the dehulled faba beans. $X$ is the moisture content, $t$ is the drying time, and $dX/dt$ represents the drying rate. The drying rate has a maximum value shortly after the start of drying [26].

### 2.2. Soaking

Faba beans were soaked in distilled water, acid solutions (0.1% citric acid at a pH of 3), and alkali solutions (1% $NaHCO_3$ at a pH of 8) for 12 h and under different temperatures (room temperature and 37 °C).

### 2.3. Microwaving

Faba bean samples were soaked as previously described, then subsequently drained, gently wiped off with a kitchen towel to remove excess water, and then microwaved at different times (30 s, 1 min, and 2 min) using an Anko microwave oven (Model P70B20AP-ST, Anko, Australia, Sydney) and applying a frequency of 2450 MHz.

### 2.4. Phytic Acid Test

A sensitive method from Haug et al. [27] was used in this study, as described in Bhinder et al. [28]. Briefly, 0.5 g of faba bean flour was stirred in a 0.2 N solution in 25 mL of HCl for three hours and then filtered. A ferric solution was prepared by dissolving 0.2 g of ammonium iron (III) sulphate in 100 mL of 2N HCl and diluting it to 1000 mL with distilled water. A 2,2′-bipyridine solution was prepared by dissolving 10 g of 2,2′-bipyridine and 10 mL of thioglycolic acid in 1000 mL distilled water. A sample extract (0.5 mL) and 1 mL of a ferric solution was mixed in a boiling water bath for 30 min. A 2,2′-bipyridine solution (2 mL) was added after the solution had cooled down, and the absorbance was measured at 519 nm by using a UV-Vis spectrophotometer (Cary 60 Instrument, Agilent, Santa Clara, CA, USA) following a reaction time of 1 min. Samples were also analysed in triplicate.

### 2.5. Fourier Transform Infrared (FTIR) Spectroscopy

Faba beans were ground into a flour using a coffee grinder. This experiment utilized Attenuated Total Reflectance (ATR) spectra (Spectrum 400, Perkin Elmer, Shelton, CT, USA). FTIR spectra were collected at a resolution of 4 $cm^{-1}$ with 32 scans in the wavelength range from 2000–400 $cm^{-1}$. The software OMNIC 8.2 was used to analyse the collected data for further data processing. First, $H_2O$ interference was subtracted in the calculation process. Then, the curve was smoothed, and the Gaussian deconvolution was calculated. By calculating each peak area, the secondary structures of the proteins in the amide regions were quantified. Samples were analysed in triplicate.

### 2.6. In Vitro Protein Digestibility

The method was developed from Hsu et al., [29], which was previously used to measure the protein digestibility for lentils [30]. In order to pass through an 80-mesh screen, all faba bean samples used for the in vitro digestion study were ground to a fine flour. A 10 mL volume of aqueous protein suspension (6.25 mg of protein/mL) in distilled

water was adjusted to a pH of 8 (with 0.1 N HCl/NaOH). This process was performed while stirring the solution at 37 °C in a water bath. To initiate digestion, a pH-adjusted multi-enzyme solution (1.6 mg/mL o trypsin, 3.1 mg/mL of chymotrypsin, and 1.3 mg/mL of peptidase) was added. An aliquot (1 mL) of this multi-enzyme solution was then added to the protein suspension whilst stirring at 37 °C. Samples were analysed in triplicate.

Protein digestibility was calculated as follows (Equation (3)):

$$\text{Digestibility (\%)} = 210.46 - 18.1x \tag{3}$$

where $x$ is the pH at a time of 10 min. The pH-drop method estimates protein digestibility by measuring the change in pH after a specified period (10 min) of hydrolysis. It is based on the principle that hydrolysis results in the release of carboxyl (-COO$^-$) and amino (-NH$_3{}^+$) groups. At neutral and alkali pHs, the free amino groups deionize, and protons (H$^+$) are liberated. The free H$^+$ groups released into the surrounding reaction medium cause a decrease in pH, and the drop in pH is recorded automatically over a 10 min period using a recording pH meter.

### 2.7. Scanning Electron Microscopy (SEM)

For sample preparation, dehulled faba beans after different treatments were used in this study. A scalpel was used to slice the faba bean samples. Then, the faba bean slices were prepared by using double-sided carbon tape glued to an aluminium stub. Since faba beans are non-conductive materials, a sputter coater was used to coat the faba bean slices with Pd/Au. Then, electron micrographs were obtained using a Zeiss ULTRA plus (Carl Zeiss SMT AG, Oberkochen, Germany) Scanning Electron Microscope (SEM) in backscatter electron detector (BSD) mode with an operating vacuum of 1 Pa (absolute). A 200–30,000× magnification was used in all images.

## 3. Results and Discussion

### 3.1. Fluidized Bed Drying

This study analysed the fluidized bed drying curve of dehulled faba beans. This section describes the effects on phytic acid and protein conformation modifications at the high constant fluidized bed drying temperature conditions of 120 °C and 140 °C.

#### 3.1.1. Drying Curve

The drying curve is an intuitive technique for calculating drying rates and times. The moisture content–time curves in Figure 2 were obtained by sampling dehulled faba beans every 2 min to measure the moisture content of faba beans. Figure 2 shows the moisture content–time curves at 120 °C and 140 °C. The marker is the measured moisture content, and the curve was obtained from the fitting equation (Equation (1)). To reach the target moisture content (12%), the process at an inlet air temperature of 120 °C took 15 min. For the inlet air temperature of 140 °C, 10 min was required to reach the target moisture content. At an inlet air temperature of 120 °C, the drying rate was slower than at the inlet air temperature of 140 °C. As expected, the drying rate increased with the increasing temperature. Changing the inlet air temperature only changed the drying rate and had no effect on the shape of the drying curve. This result is consistent with the findings of Cheng et al. [10] for chickpeas.

Figure 3 shows the predicted temperatures of the faba beans during the fluidized bed drying process at 120 °C and 140 °C, as obtained from Equation (2). According to Keey [27], during the drying process, the evaporation of moisture in the material being dried cools the material, and the drying rate decreases as drying proceeds. Therefore, the temperatures of the faba beans increase during drying. The temperatures of the faba beans are also higher at larger air temperatures. At the target moisture content (12%), the temperatures of the faba beans at an air temperature of 140 °C for 10 min were predicted to be 110 °C, and the temperatures of the faba beans at an air temperature of 120 °C for 15 min were predicted to be 102 °C.

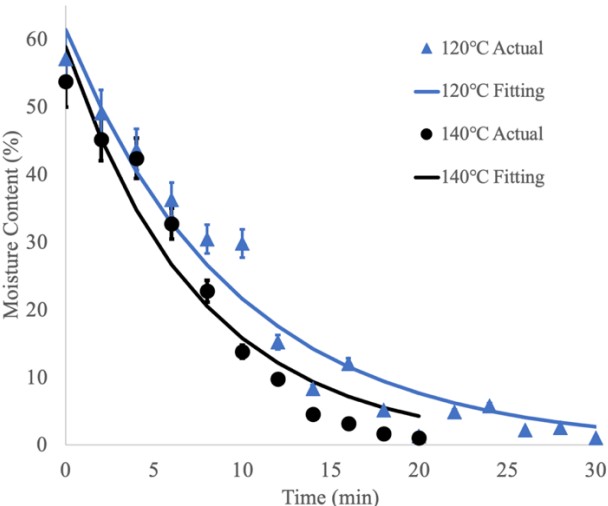

**Figure 2.** Moisture content–time curves for the fluidized drying process of dehulled faba beans at 120 °C and 140 °C.

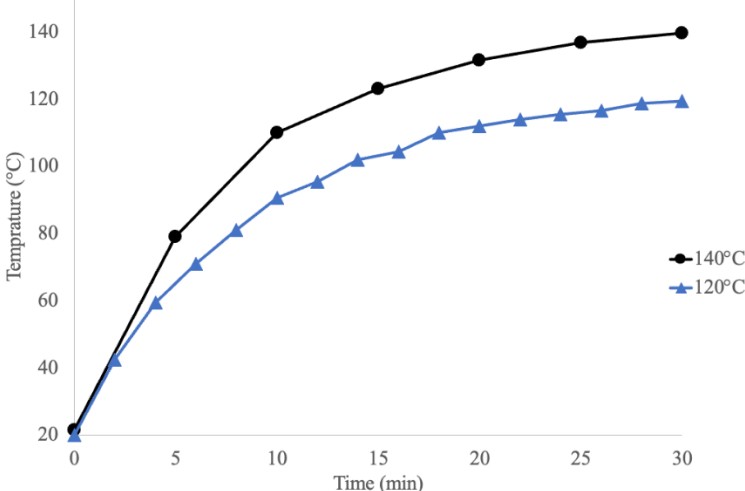

**Figure 3.** Predicted temperature–time curves for dehulled faba beans during fluidized bed drying at 120 °C and 140 °C.

3.1.2. Phytic Acid

The phytic acid content in broad beans can negatively affect the bioavailability of minerals and have an inhibitory effect on protein absorption [31]. The content of phytic acid in raw dehulled faba beans is around 710 mg/100 g on a dry basis. This range of concentrations for phytic acid is reasonable. The values reported by Dhull et al. (2022) also supports this value [32]. For instance, the level of phytic acid in untreated faba beans here was 8.36 mg/g, which is in the same range. Further, in this work, dehulled faba beans were used. Dehulling is a common form of processing for pulses, in which the seed hull is separated from the cotyledon. Nutrient compositions and the level of anti-nutritional factors are also affected by the dehulling process [23].

Tables 2 and 3 show the changes in phytic acid concentrations over 5 min intervals from 120 °C and 140 °C fluidized bed drying. At the target moisture content (12%), 120 °C fluidized bed drying for around 15 min reduced the phytic acid content by about 15 ± 2%. In addition, 8 ± 3% of the phytic acid content was reduced by drying in the fluidized bed at 140 °C for 10 min. Comparing the concentrations at the target moisture content from the drying temperatures of 120 °C and 140 °C, the phytic acid content did not decrease with an increase in the drying temperature. As shown in Table 2, with the extension of the drying time, the content of phytic acid showed a downward trend, and the content

of phytic acid decreased to a limited extent. However, this trend is not significant at the inlet air temperature of 140 °C. This means that fluidized bed drying did not play a large role in reducing the phytic acid content. These results are consistent with the literature, where it has been found that the content of phytic acid does not decrease with increases in the drying time and drying temperature, so phytic acid is more stable than trypsin inhibitors during the drying process [23]. Daneluti et al. [32] found that phytic acid thermally decomposes at a temperature of 380 °C, resulting in the reduction of carbon and hydrogen. This means phytic acid is thermally stable during this drying treatment. Hence, increasing the temperature during drying does not appear to be an optimal way to remove phytic acid.

**Table 2.** The effects of different drying times on phytic acid reduction for fluidized bed drying at 120 °C.

| Drying Time (min) | Moisture Content (%) | Phytic Acid (mg/100 g) | Reduction (%) |
|---|---|---|---|
| 5 | 40 | 770 ± 40 | ≤0 |
| 10 | 22 | 656 ± 24 | 8 ± 3 |
| 15 | 10 | 603 ± 16 | 15 ± 2 |
| 20 | 8 | 562 ± 64 | 21 ± 9 |

**Table 3.** The effects of different drying times on phytic acid reduction for fluidized bed drying at 140 °C.

| Drying Time (min) | Moisture Content (%) | Phytic Acid (mg/100 g) | Reduction (%) |
|---|---|---|---|
| 5 | 31 | 572 ± 33 | 19 ± 5 |
| 10 | 14 | 655 ± 47 | 8 ± 7 |
| 15 | 8 | 548 ± 41 | 22 ± 6 |
| 20 | 4 | 814 ± 168 | ≤0 |

### 3.2. Soaking

This study analysed the effects of different soaking conditions on phytic acid concentrations. This section compares the change in the phytic acid levels after soaking in acidic, alkaline, and neutral solutions for 12 h and the change in the phytic acid levels at different temperatures (RT and 37 °C).

Soaking is the most popular processing method to remove phytic acid, or to reduce the phytate content, of pulses [33]. According to Table 4, soaking can help to reduce the phytic acid content to a certain extent. After soaking for 12 h, 10–50% of the phytic acid was removed under different conditions. In particular, soaking for 12 h under acidic conditions at 37 °C removed 51 ± 11% of the phytic acid in this study. According to the research of Osman et al. [34], soaking goat peas in water for 6 h at RT reduced phytic acid concentrations by 3.9%. Vidal-Valverde et al. [14] found that soaking lentils in 0.1% citric acid at room temperature reduced phytic acid concentrations by 37%. The results obtained in this experiment are slightly higher than those in the literature, possibly because dehulled faba beans were used in this experiment. The seed coat of pulses is likely to resist water penetration during soaking.

Soaking can remove part of the phytic acid, but the reduction may not be complete. The soaking process may play a small role in removing phytic acid due to the following reasons. Soaking pulse in water results in the release of water-soluble phytic acid or phytate (sodium or potassium salt) [33,35]. During the soaking process, diffusion of water occurs into the pulse matrix, the phytic acid dissolves in the water, and the phytase is activated. The pulse matrix is inherently porous, and this type of structure allows for faster water diffusion and the dissolution of phytic acid in water [33]. Studies have shown that soaking for different periods of time can effectively reduce phytic acid concentrations [36–38]. The effect of soaking on phytic acid reduction depends largely on the soaking conditions,

namely the pH and temperature of the soaking solution. However, longer soaking periods can also lead to the loss of soluble proteins [39].

**Table 4.** The effects of different soaking conditions on phytic acid.

| Pre-Treatment | Moisture Content (%) | Phytic Acid (mg/100 g) | Reduction (%) |
|---|---|---|---|
| Water 12 h | 55 | 436 ± 75 | 39 ± 11 |
| Water 12 h 37 °C | 57 | 605 ± 68 | 15 ± 9 |
| Acid 12 h * | 56 | 538 ± 68 | 24 ± 10 |
| Acid 12 h 37 °C | 55 | 349 ± 75 | 51 ± 11 |
| Alkali 12 h ** | 54 | 552 ± 49 | 22 ± 7 |
| Alkali 12 h 37 °C | 55 | 640 ± 45 | 10 ± 6 |

* Acid soaking (0.1% citric acid, a pH of 3). ** Alkali soaking (1% $NaHCO_3$, a pH of 8).

### 3.3. Microwaving

Applications of microwaves in food processing have been reported and studied, such as in thawing, blanching, baking, drying, pasteurization, sterilization, and the extraction of biologically active compounds [16,40,41]. Table 5 shows the effect of different microwaving conditions on phytic acid concentrations. In this study, dehulled faba beans were soaked in neutral, acid, and alkali solutions before being subjected to microwaving. Then, they were microwaved for 30 s, 1 min, and 2 min. As the microwave time increased, the phytic acid content decreased. This trend was most pronounced for acidic soaking conditions. After 2 min of microwaving, most of the phytic acid was removed. In other words, microwaving removed phytic acid from the dehulled faba beans. According to the study of Suhag et al. [16], microwaves can help reduce anti-nutritional compounds present in foods, thereby improving in vitro protein digestibility and the safety and quality of food grains. This conclusion is also consistent with our results.

**Table 5.** The effects of different microwaving conditions on phytic acid.

| Pre-Treatment | Moisture Content (%) | Phytic Acid (mg/100 g) | Reduction (%) |
|---|---|---|---|
| Water 12 h + 30 s M | 48 | 271 ± 98 | 62 ± 14 |
| Water 12 h + 1 min M | 37 | 314 ± 112 | 56 ± 16 |
| Water 12 h + 2 min M | 13 | ≤0 | 100 ± 2 |
| Acid 12 h * + 30 s M | 52 | 506 ± 180 | 29 ± 25 |
| Acid 12 h * + 1 min M | 31 | 257 ± 85 | 64 ± 12 |
| Acid 12 h * + 2 min M | 6 | 29 ± 1 | 96 ± 1 |
| Alkali 12 h ** + 2 min M | 12 | 207 ± 9 | 71 ± 9 |
| Alkali 12 h (37 °C) ** + 2 min M | 16 | ≤0 | 100 ± 1 |

M (microwaving). * Acid soaking (0.1% citric acid, a pH of 3). **Alkali soaking (1% $NaHCO_3$, a pH of 8).

The temperature of faba beans after microwaving was measured by using an alcohol-in-glass thermometer (with a precision of ±2 °C). The temperatures after 30 s, 1 min, and 2 min of microwaving were 77 ± 4, 82 ± 5, and 92 ± 3 °C, respectively. Compared with the fluidized bed drying process at 120 °C, microwaving takes two minutes to reach a temperature of 92 °C. This means the temperature of the faba beans increased very quickly. During the 2 min of microwaving, the water and the beans were close to the boiling point of water at atmospheric pressure (100 °C). In addition, the initial moisture content of faba beans after soaking is around 55%. The microwaving process caused a pressure difference between the water and the solid faba beans. At the same time, the water evaporated quickly. Thus, the internal structure of the faba bean may have changed during the microwaving process, and this change was confirmed by SEM images, shown in Section 3.4.

Timber shows similar behaviours in the microwave drying process [42]. For example, Terziev el al. [42] reported that in timber drying, the advantage of microwaving is that it can dry timber faster than conventional drying methods and can preserve quality. The microwaving process changes wood structures, such as pit openings, by rupturing the pit

membranes or weakening the middle lamella in timber cell walls. This change is due to microwave heating being concentrated in the wet regions of moist wood and evaporating water and creating pressure gradients between cells [42,43]. This process also happens in faba bean microwaving, which assists in the reduction of phytic acid levels. The next section on scanning electron microscopy (SEM) also illustrates the structural changes in faba beans during the microwaving process.

### 3.4. Scanning Electron Microscopy

Figure 4a,b show the SEM images of the raw faba beans at different magnifications. Figure 4c,d show the structure of the faba beans after soaking. These images are all characteristic of the internal structure of faba bean slices. Under high magnification, Figure 4a shows the starches, proteins, and cell walls in faba beans. In the dry condition (raw faba bean), the intercellular spaces are large, as shown in Figure 4b. After soaking, the cells tended to expand significantly, and the sizes of the intercellular spaces decreased.

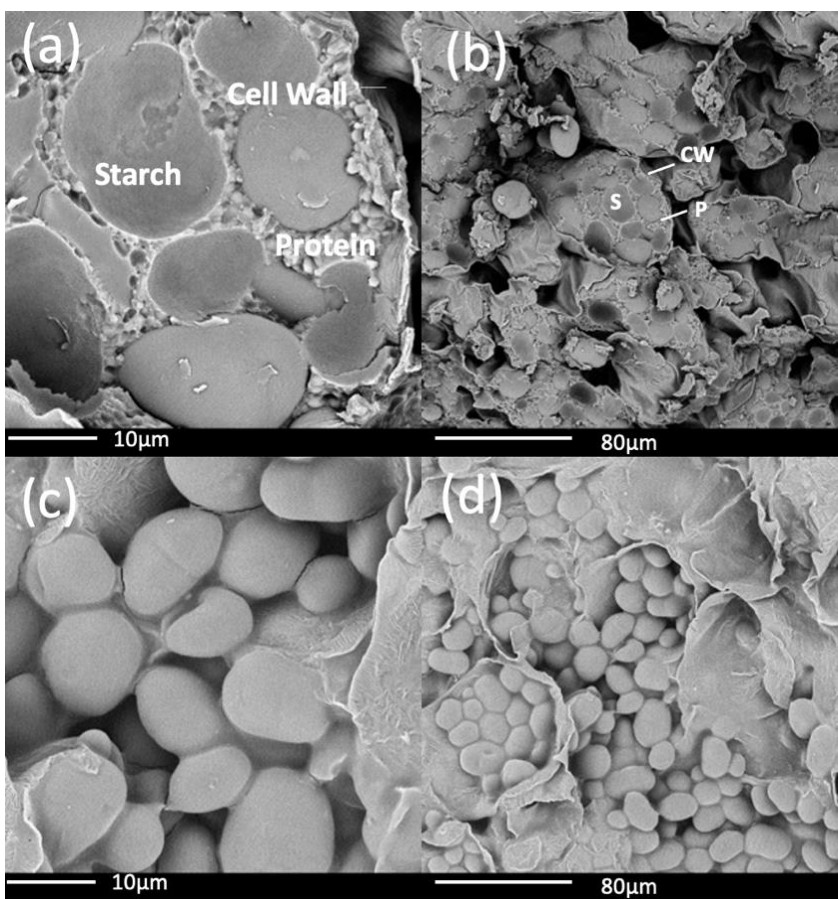

**Figure 4.** SEM images, slices of faba beans at different magnifications ($1000\times-5000\times$): (**a**) raw faba beans (dehulled), $5000\times$; (**b**) raw faba beans (dehulled), $1000\times$; (**c**) soaking in water for 12 h, $3000\times$; (**d**) soaking in water for 12 h, $1000\times$; the structure shows starch (S), protein (P), and cell wall (CW).

Figure 5a,b show the SEM images after microwaving at different magnifications. The cell walls of the faba beans became damaged after microwaving. Starch and protein have been lost in some cells. These losses are consistent with the significant changes in the protein secondary structure after microwaving. Compared with the raw faba beans, the structure inside the cells after microwaving does not appear to be so restricted. This change may be a significant reason for the reduction in phytic acid levels.

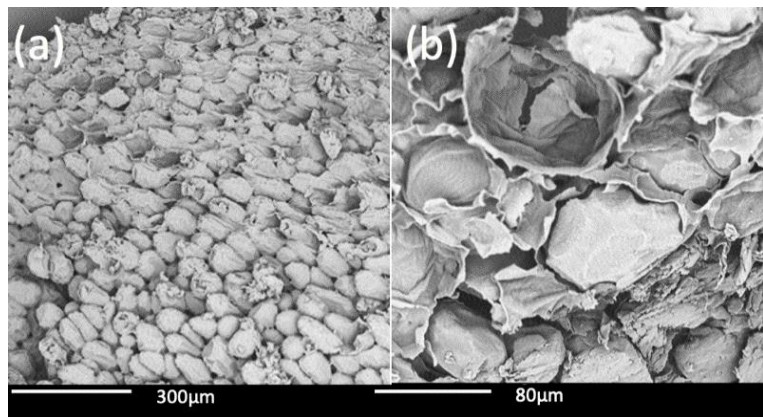

**Figure 5.** SEM images, slices of faba beans after microwaving at different magnifications (255×–5000×): (**a**) dehulled faba beans after 12 h of soaking in water and 30 s of microwaving, 255× and (**b**) dehulled faba beans after 12 h of soaking in water and 2 min of microwaving, 1000×.

### 3.5. Protein Conformation Modification

In the Fourier transform infrared (FTIR) spectrum, there are typical protein bands: Amide I (1600–1700 cm$^{-1}$), Amide II (1500–1580 cm$^{-1}$), and Amide III (1200–1400 cm$^{-1}$). Amide I delineates the strongest vibrational mode and is important for revealing and analysing protein secondary structures [44]. In this study, the Gaussian deconvolution analysis was carried out for the amide I region. This analysis helps to quantify changes in protein secondary structure. In the amide I region, protein secondary structures include β-sheets (1600–1638 cm$^{-1}$), unordered structures (1638–1650 cm$^{-1}$), α-helices (1650–1600 cm$^{-1}$), β-turns (1660–1680 cm$^{-1}$), antiparallel β-sheets (1680–1688 cm$^{-1}$), and β-type structures (1690–1695 cm$^{-1}$) [10,45].

Figure 6a shows the amide I spectrum of the raw faba beans with its Gaussian spectral deconvolution. As shown in Figure 6, raw faba beans have the most peaks, which were neatly arranged. After different treatments, the secondary structure of the protein was changed. Especially after soaking (Figure 6d) and microwave treatment (Figure 6e), the number and shape of the peaks showed significant changes.

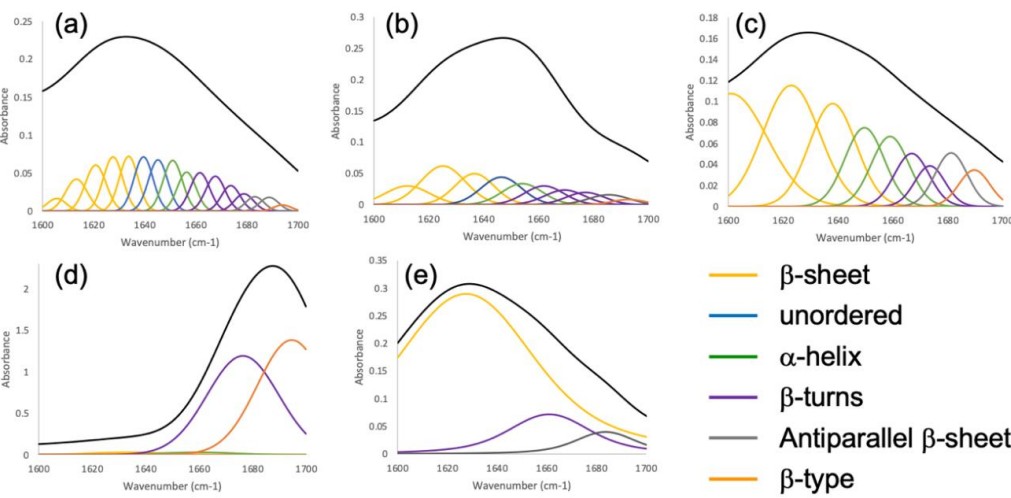

**Figure 6.** FTIR spectrum showing amide I bands of faba bean powder. The outer envelope is the original spectrum, and the individual component peaks underneath are the results of regression analysis. The peaks are associated with different secondary structures. (**a**) Raw faba bean; (**b**) 120 °C fluidized bed drying for 15 min after soaking; (**c**) 140 °C drying for 10 min after soaking; (**d**) soaking for 12 h (undried); and (**e**) microwaving for 2 min (soaked, not dried in the fluidized bed, but having a moisture content of 5–7%).

As an example, sixteen Gaussian bands were resolved for the raw faba beans (Figure 6a), centred at 1606, 1613, 1621, 1628, 1634, 1639, 1645, 1651, 1656, 1662, 1667, 1674, 1679, 1683, 1689, and 1694 cm$^{-1}$. Among them, the peaks at 1600–1638 cm$^{-1}$ were considered to represent the secondary structure of β-sheets. The peaks at 1639 and 1645 cm$^{-1}$ were assigned to unordered structures. The peaks at 1651 and 1656 cm$^{-1}$ were interpreted as a-helices. The wavebands from 1662–1679 cm$^{-1}$ were considered to be β-turns. The peaks at 1683 and 1689 cm$^{-1}$ were considered to be antiparallel β-sheets. The peak at 1694 cm$^{-1}$ was interpreted as a β-type structure. In Figure 6, peaks of the same colour represent the same protein secondary structure. By calculating the peak area, the proportion of different protein secondary structures can be obtained. The relative percentages of different structural components in the secondary structure of the raw faba beans are shown in Table 6.

**Table 6.** The relative percentages of different structural components in the secondary structure of raw faba beans, fluidized bed dried at 120 °C and 140 °C after soaking, soaked faba beans (undried), and microwaved faba beans (soaked, not dried in the fluidized bed, but having a moisture content of 5–7%).

| Secondary Structure Component | Frequency (cm$^{-1}$) | Raw Faba Beans (%) | FBD 120 °C_ 15 min (%) | FBD 140 °C_ 10 min (%) | Soaking for 12 h (%) | Microwaving for 2 min (%) |
|---|---|---|---|---|---|---|
| β-sheets | 1600–1638 | 42 | 46 | 58 | 1 | 79 |
| Unordered | 1638–1650 | 18 | 14 | 0 | 0 | 0 |
| α-helices | 1650–1660 | 15 | 11 | 21 | 1 | 0 |
| β-turns | 1660–1680 | 20 | 22 | 11 | 55 | 15 |
| Antiparallel β-sheets | 1680–1688 | 5 | 5 | 6 | 0 | 6 |
| β-type structures | 1690–1695 | 1 | 2 | 4 | 42 | 0 |
| σ (standard error, compared with raw faba bean) | | | 0.08 | 0.27 | 0.71 | 0.45 |

By comparing with the protein secondary structure in the raw faba beans, we can observe the effect of different treatments on the protein secondary structures. For the raw faba beans, the β-sheet content was 42%, and after fluidized bed drying and microwaving, the content of β-sheets increased. In comparison, the content of the β-sheets decreased to 1% after another 12 h of soaking. By comparing the standard error with the raw faba beans, it was observed that in the fluidized bed drying process, when the target moisture content is present, the higher the drying temperature, the greater the change in the secondary structure of the protein. This conclusion is also consistent with the results of Cheng et al. [10]. On the other hand, soaking had the greatest impact on the secondary structure of proteins among the three different treatments; this may be due to the high moisture content (55 ± 5%) of the faba bean samples after soaking. Also, applying 2 min of microwaving had an impact on the protein secondary structure.

### 3.6. In Vitro Protein Digestibility

This study analysed the changes in protein digestibility under different treatment conditions. All the treatments in this experiment were based on dehulled faba beans. For the fresh faba bean seeds with hulls, the in vitro protein digestibility was 75.5%. For fresh faba beans, regardless of whether they were dehulled, treated in a microwave and dehulled, soaked and dehulled, or fluidized bed dried and dehulled, the in vitro protein digestibility was significantly improved. Among them, the in vitro digestibility of protein is the highest after 12 h of water soaking and 2 min of microwave processing, at a value of 88.3% (Table 7).

**Table 7.** The effects of different pre-treatment conditions on in vitro protein digestibility.

|   | Treatment | pH | Digestibility (%) | Increase (%) |
|---|---|---|---|---|
| 1 | Fresh faba bean (Hulled) | 7.5 | 75.5 ± 0.5 | |
| 2 | Raw faba beans (Dehulled) | 6.8 | 87.4 ± 1.6 | 15.7 |
| 3 | FBD 140 °C for 10 min (Dehulled) | 6.8 | 88.0 ± 1.5 | 16.5 |
| 4 | FBD 120 °C for 15 min (Dehulled) | 6.8 | 88.0 ± 0.5 | 16.4 |
| 5 | Water Soaking for 12 h + Microwaving for 30 s (Dehulled) | 6.9 | 86.4 ± 0.7 | 14.4 |
| 6 | Water Soaking for 12 h + Microwaving for 2 min (Dehulled) | 6.8 | 88.3 ± 0.6 | 16.8 |
| 7 | Soaking for 12 h (Dehulled) | 7.0 | 84.9 ± 0.2 | 12.4 |

These results are consistent with those of Luo et al. [46]. According to that study [46], the in vitro protein digestibility of raw seed was 72.7%. The dehulling process increased the digestibility by 3.8%. Soaking increased the digestibility by 1.2–2.6%. This result means that the dehulling process had a positive effect on protein digestibility. The soaking process had little effect on protein digestibility. The digestibility of combining microwaving, soaking, and dehulling reached 82.3%. There are many factors that affect pulse protein in vitro digestibility, such as anti-nutritional factors (phytic acid and tannins), and thermal treatment [47]. For the fresh faba beans (hulled), the content of phytic acid was around 980 mg/100 g [13]. This result was much higher than the content of phytic acid in the raw faba beans (dehulled) (710 mg/100 g). Compared with the fresh faba beans (hulled), the in vitro protein digestibility of the raw faba beans increased by 11.9%. This result means that the decrease in the phytic acid content appears to improve in vitro protein digestibility.

*3.7. Overall Discussion*

Overall, microwaving was the most effective process for removing phytic acid, comparing with fluidized bed drying and soaking. Microwaving for 2 min removed most of the phytic acid. For improving in vitro protein digestibility, dehulling was an effective method. The dehulled faba beans after microwaving and fluidized bed drying also had high in vitro protein digestibility, since microwave processing changes the physical structure of dehulled faba beans, but microwave processing also affects the protein secondary structure.

Nevertheless, there is a negative correlation between the phytic acid content and the glycaemic index [48]. Phytic acid may not need to be completely removed, and certain amounts of phytic acid can reduce the glycaemic index. This study provides different pre-treatment methods to meet the needs for different levels of phytic acid.

Furthermore, by comparing the phytic acid levels in raw faba beans, raw chickpeas, raw lentils, and raw mung beans, the levels in faba beans are twice those in the other three pulses [7]. The phytic acid levels in raw chickpeas, raw lentils, and raw mung beans are between 580–1250 mg/100 g pulses [7]. In comparison, the phytic acid levels in raw faba beans are 640–2170 mg/100 g pulses [7]. The treatments used in this study are likely to work well for pulses with high phytic acid contents, and these methods should be broadly applicable to other pulses, including chickpeas, lentils, mung beans, lupins, and field peas.

**4. Conclusions**

This paper studied the effects of different treatments for dehulled faba beans on the reduction of phytic acid concentrations in these beans. The treatments applied were fluidized bed drying, soaking, and microwaving. Fluidized bed drying did not reduce the concentration of phytic acid from faba beans significantly, suggesting that thermal treatment had little effect on phytic acid levels. Soaking reduced phytic acid levels to a certain extent, and microwaving had a higher effect on reducing phytic acid levels in the faba beans. Different soaking conditions and different microwaving times were assessed in this study. Soaking in water, acid, and alkali removed 39 ± 11%, 24 ± 9 and 51 ± 11% of phytic acid, respectively. This result shows that different soaking environments have an impact on the removal of phytic acid. Using microwaving for 2 min, under different soaking conditions, the phytic acid was removed by over 96%.

This study used SEM and FTIR to analyse the physical structure of faba beans under different treatment conditions and the secondary structures of proteins in faba beans. The physical structure of the faba bean and the secondary structure of the protein changed after microwaving, which may connect with the reduction in phytic acid levels. The in vitro protein digestibility tests with different treatments showed that dehulling faba beans improved the digestibility of the proteins. On the other hand, after microwaving, the digestibility of the protein also showed some improvements. In general, the results show that microwaving is a potential pulse protein processing technology, because it not only reduces the content of anti-nutritional factors, but it also improves the digestibility of proteins.

**Author Contributions:** Conceptualization, S.C. and T.A.G.L.; methodology, T.A.G.L. and S.C.; validation, S.C., T.A.G.L., D.J.S., C.W. and V.M.; formal analysis, S.C. and T.A.G.L.; investigation, S.C.; resources, T.A.G.L.; data curation, T.A.G.L. and S.C.; writing—original draft preparation, S.C. and T.A.G.L.; writing—review and editing, S.C., T.A.G.L., D.J.S., C.W. and V.M.; visualization, T.A.G.L. and S.C.; supervision, T.A.G.L. and D.J.S.; project administration, T.A.G.L.; funding acquisition, T.A.G.L. All authors have read and agreed to the published version of the manuscript.

**Funding:** Global Innovation Linkages Program Round 3, Transitioning Australian pulses into protein-based food industries, no grant number.

**Data Availability Statement:** Data are contained within the article.

**Conflicts of Interest:** Author Daniel Skylas and Chris Whiteway were employed by AEGIC. The remaining authors declare that the research was conducted in the absence of any commercial or financial relationships that could be construed as a potential conflict of interest.

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
