# Peer review of "The Effects of Fluidized Bed Drying, Soaking, and Microwaving on the Phytic Acid Content, Protein Structure, and Digestibility of Dehulled Faba Beans"

_processes, doi:10.3390/pr11123401_

Round 1

Reviewer 1 Report

Comments and Suggestions for Authors

The submitted article examined the influence of microwaving on phytic acid  levels in faba beans and the changes in the protein structure of faba beans and the changes in the cell structure of faba beans during microwaving. The article is correctly written. Some specific comments are given in the uploaded file.

Author Response

Thank you for your valuable comments and suggestions, which are greatly appreciated. We incorporated our manuscript based on your suggestions. 

Reviewer 2 Report

Comments and Suggestions for Authors

Development of methods for minimizing/elimination of anti-nutritional components has high relevance for the industry practice, as well. As the authors mentioned, phytic acid is one of the main antinutritive component in faba beans. However the manuscript contains some interesting information, but there are many unclear parts and aspects, furthermore, the ‘storyline’ is unclear, as well.

Some comments:

Abstract summarize the main essence of the study. Therefore, in my opinion, the keywords should be consistent with the content of the Abstract, as well. However, ’Fluidized bed drying’ is not mentioned in the Abstract.

There cannot be found data and information in the Introduction section why has the microwave irradiation stronger effect on antinutritive compounds- phytic acid than conventional thermal treatment. Please discus briefly this topic, as well.

The title of the manuscript suggests a topic related to the applicability and efficiency of different methods for the removal/minimization of phytic acid content of faba beans. In Results and discussion section ’3.1.2. Phytic Acid’ discuss the change of phytic acid content during drying, section 3.2 during soaking, section 3.3 during microwaving. Therefore, in my opinion, section 3.1.1. ’ Drying Curve and Simulation Modelling’ contains unnecessary and irrelevant data and information considering the main focus of the study.

Tables do not contain standard deviation/measuring errors. Therefore, the significance of the differences cannot be established.

In section 3.1.2.  there can not be found data and information related to the temperature (and temperature variation/change) of beans (temperature data presented just for drying media).

Establishment (in present form with present content) in line 265 ’Soaking can remove part of the phytic acid, but the reduction may not be significant’ contradicts to sentence in line 271-273, and data presented in Table 4.

Manuscript does not presented temperature of processed material during microwave treatment. In my opinion, therefore, the efficiency of drying (at given temperatures) is not comparable with MW method.

The differences’ physical structure of faba bean using different treatment methods should cause by the different final temperature and/or different temperature ramp, for instance. Another aspect: the different dehydration ratio (and final moisture content) can also cause differences in physicochemical parameters. These aspects should be cleared, as well.

In my opinion, the microbial parameters should be investigated as well (mainly if soaking is applied).

The effects of treatment on other compounds (and ‘technofunctional parameters as well) which has importance for the further utilization technologies/methods and/or product quality and safety.

In my opinion, sentences in line 78-78 should be moved to after line 43.

Please check and correct the symbols unit in the manuscript (see line 99: ’80oC’, line 162 for instance).

Please correct Eq.3.

Author Response

Thank you for your valuable comments and suggestions, which are greatly appreciated. Please see the attachment.

Reviewer 3 Report

Comments and Suggestions for Authors

This paper has studied the effect of different treatments, i.e. fluidized bed drying, soaking and microwaving for dehulled faba beans on the reduction of phytic acid concentrations in these beans. The results found that microwaving had a higher effect in reducing phytic acid levels in the faba beans.

This study used SEM and FTIR to analyse the physical structure of faba beans under different treatment conditions and the secondary structures of proteins in faba beans.

The topic is food engineers. The topic fits the scope of the journal Processes.

However, the manuscript have some drawbacks need to be revised.

1.       Introduction part is too general. In introduction part, the research status or specific problems with regard to the fluidized bed drying, soaking and microwaving methods of dehulled faba beans on the reduction of phytic acid concentrations may be reviewed. Not the general descriptions on microwaving applications.

2.       What is the fluidized situation in the fluidized bed drying? I mean from fluidization point of view, the authors have found this method did not reduce the concentration of phytic acid from faba beans significantly. Would the fluidization or higher temperature works?

3.       What the information of SEM figure offers? How did the sample making for SEM? This was not mentioned in the manuscript.

4.       Detailed analysis on FTIR spectrum result in figure 7 should be given. In this figure the fluidizing bed drying temperature is 120 degree Celsius but in the other part, the temperature is 80 degree Celsius, which one is correct?

5.       How to apply the results of the present manuscript to some other beans?

6.       A slight discussion on the results may be good for readers to get the whole picture of this manuscript and conclusions.

7.       Overall, this manuscript is somehow like a technical report not scientific paper. The studied parameters, mechanisms, proof for results should be given. The abstract part should be revised and the motivation and studied parameters should be highlighted.

Author Response

Thanks for the helpful comments. Please see the attachment.

Round 2

Reviewer 2 Report

Comments and Suggestions for Authors

The manuscript has an interesting topic. Authors have revised the mnauscript thoroughly according to reviewers comments ad suggestions and provided detailed answers for reviewers questions. The overall scientific quality has been improved significantly due to the reision. I agree and accept all modifications made by the authors.

Comments, suggestions:

Please check the typos in the whole manucript (for example:see Celsius signs etc.).

Please check the reference in the text. See line524-527.

Author Response

Thanks for this comments. Please see the attachment.